# Treatment outcomes in HIV infected patients older than 50 years attending an HIV clinic in Harare, Zimbabwe: A cohort study

**Tinei Shamu**[1,2,3]*, **Cleophas Chimbetete**[1], **Matthias Egger**[2,4,5], **Tinashe Mudzviti**[1,6]

**1** Newlands Clinic, Newlands, Harare, Zimbabwe, **2** Institute of Social and Preventive Medicine, University of Bern, Bern, Switzerland, **3** Graduate School of Health Sciences, University of Bern, Bern, Switzerland, **4** Centre for Infectious Disease Epidemiology and Research, School of Public Health and Family Medicine, University of Cape Town, Rondebosch, Western Cape, South Africa, **5** Population Health Sciences, Bristol Medical School, University of Bristol, Bristol, United Kingdom, **6** School of Pharmacy, College of Health Sciences, University of Zimbabwe, Harare, Zimbabwe

* tineis@newlandsclinic.co.zw

**Data Availability Statement:** https://doi.org/10.6084/m9.figshare.14525487.

**Funding:** The authors received no specific funding for this work.

## Abstract

There is a growing number of older people living with HIV (OPLHIV). While a significant proportion of this population are adults growing into old age with HIV, there are also new infections among OPLHIV. There is a lack of data describing the outcomes of OPLHIV who commenced antiretroviral therapy (ART) after the age of 50 years in sub-Saharan Africa. We conducted a cohort study of patients who enrolled in care at Newlands Clinic in Harare, Zimbabwe, at ages ≥50 years between February 2004 and March 2020. We examined demographic characteristics, attrition, viral suppression, immunological and clinical outcomes. Specifically, we described prevalent and incident HIV-related communicable and non-communicable comorbidities. We calculated frequencies, medians, interquartile ranges (IQR), and proportions; and used Cox proportional hazards models to identify risk factors associated with death. We included 420 (57% female) who commenced ART and were followed up for a median of 5.6 years (IQR 2.4–9.9). Most of the men were married (n = 152/179, 85%) whereas women were mostly widowed (n = 125/241, 51.9%). Forty per cent (n = 167) had WHO stage 3 or 4 conditions at ART baseline. Hypertension prevalence was 15% (n = 61) at baseline, and a further 27% (n = 112) had incident hypertension during follow-up. During follow-up, 300 (71%) were retained in care, 88 (21%) died, 17 (4%) were lost to follow-up, and 15 (4%) were transferred out. Of those in care, 283 (94%) had viral loads <50 copies/ml, and 10 had viral loads >1000 copies/ml. Seven patients (1.7%) were switched to second line ART during follow-up and none were switched to third-line. Higher baseline CD4 T-cell counts were protective against mortality (p = 0.001) while male sex (aHR: 2.29, 95% CI: 1.21–4.33), being unmarried (aHR: 2.06, 95%CI: 1.13–3.78), and being unemployed (aHR: 2.01, 95%CI: 1.2–3.37) were independent independent risk factors of mortality. There was high retention in care and virologic suppression in this cohort of OPLHIV. Hypertension was a common comorbidity. Being unmarried or unemployed were significant predictors of mortality highlighting the importance of sociologic factors among OPLHIV, while better immune competence at ART commencement was protective against mortality.

**Competing interests:** The authors have declared that no competing interests exist.

## Introduction

The number of older people living with HIV (OPLHIV) above the age of 50 years continues to grow due to the increased life expectancy of people living with HIV. In 2019, the proportion of OPLHIV among all people living with HIV was estimated to have risen to 21% globally, from 8% in 2000 [1, 2]. Low- and middle-income countries (LMIC) have the highest numbers of OPLHIV and this number will increase further [2]: several studies suggest that the proportion of OPLHIV will more than double by 2030 [3–5]. This population includes both people with chronic HIV infection from early adulthood and new infections among older adults for reasons including lack of information on safe sex and a lower likelihood of protected sex, especially among older men, compared to younger people [6].

OPLHIV have been shown to have better virologic outcomes and adherence to antiretroviral therapy (ART) than younger people, but have higher mortality and slower immune recovery [7–9]. Compounding the age-related thymic contraction, HIV infection affects both B- and T-cell function leading to accelerated immunosenescence and consequently increased risk of non-AIDS related, age associated comorbidities in elderly patients [4, 10, 11]. Management of OPLHIV requires a multidisciplinary team approach, taking into account the onset of non-communicable diseases, premature ageing, and an increased risk of drug-to-drug interactions and drug toxicities due to polypharmacy [4]. Polypharmacy also complicates selection of ART regimens and the increased chronic pill burden may lead to treatment fatigue [12]. Several studies have shown that men have lower uptake of ART services, increased severity of HIV disease at ART initiation and higher mortality on ART in sub-Saharan Africa [13–15].

To our knowledge, there are no studies from Zimbabwe to document outcomes of ART in OPLHIV. However, there are emerging data on ART outcomes of OPLHIV in sub-Saharan Africa [16, 17]. With an estimated hypertension prevalence of 30%, a diabetes prevalence of 5%, and an increasing incidence of cancer in the Zimbabwean general population, it is imperative to study the prevalence of these non-communicable diseases among OPLHIV [18–20]. In addition, antiretrovirals that continue to be used in Zimbabwe such as abacavir, efavirenz, zidovudine and lamivudine may increase the risk of cardiovascular disease [21].

In this study, we present the sex-stratified prevalence of chronic non-communicable diseases, HIV viral suppression, immunological and clinical outcomes of older patients who registered for HIV management at an HIV clinic in Harare, Zimbabwe between February 2004, and March 2020.

## Methods

### Study design and ethics statement

This study was conducted within the ongoing International epidemiological Databases to Evaluate AIDS Southern Africa (IeDEA) cohort study of patients attending the Newlands Clinic in Harare, Zimbabwe [22, 23]. The study was approved by the Newlands Clinic Research Team and received ethical approval from the Medical Research Council of Zimbabwe (MRCZ No. A1336). Participants provided written informed consent.

### Study setting

Newlands Clinic is a referral outpatient clinic providing comprehensive HIV care which is supported by the Ruedi Luethy Foundation, a Swiss-based Private Voluntary Organization. Newlands Clinic was established in February 2004 and operates under a Public-Private Partnership with the Ministry of Health and Child Care of Zimbabwe. Apart from providing antiretroviral therapy to people living with HIV, the clinic also provides, among other services,

onsite laboratory investigations, psychosocial support, reproductive health services including cervical cancer screening, dental services, food support, and bus fare assistance. The patient population is predominantly from communities with high prevalence of poverty and unemployment in the Greater Harare area. Further details on the clinic's operations are provided elsewhere [24]. Newlands Clinic is part of the Southern African region of the International epidemiological Databases to Evaluate AIDS Southern Africa (IeDEA), an international research consortium of HIV observational databases that includes four regional networks in sub-Saharan Africa [23].

## Study procedure

We included all patients aged 50 years or more, who were ART naïve at the time of registration, and started ART at the clinic. Patients were followed from the date of ART commencement until they were transferred out, lost to follow up, or died. Follow up of patients still in care was censored on April 30, 2020. We abstracted demographic, clinical and laboratory data from patient records stored in the clinic's password protected electronic medical record. Abstracted records were anonymized by removing names and assigning sequential numeric identifiers. We defined baseline CD4 count as a CD4 count closest to the date of initiating ART within 90 days before ART commencement. HIV viral load (VL) measurements were available at the clinic from the year 2014 onwards. The most recent CD4 T-cell count and VL were defined as the last measured CD4 T-cell count or viral load for each patient. CD4 T-cell counts were measured using the CyFlow® Counter (Flow cytometer) and CD4 easy count kits. HIV viral loads were measured using the COBAS® AmpliPrep/COBAS® TaqMan® HIV-1 Test, version 2.0.

Nurses measured blood pressure during routine clinic visits after the patient had been seated for at least 5 minutes, with the arm supported at the level of the heart and elbow flexed slightly. Nurses ensured the arm was not constricted by tight clothing and placed the cuff on naked skin with the centre of the bladder over the brachial artery. Diagnosis of hypertension was made when a patient's systolic or diastolic blood pressure was above 140mm Hg or above 90mm Hg, respectively, on two or more days. Risk reduction of major adverse cardiovascular events using aspirin was only provided for patients with history of myocardial infarction. Type II diabetes mellitus was diagnosed when a patient had a random plasma glucose $> 11$mmol/L or a fasting plasma glucose $\geq 7$mmol/L. We grouped all forms of chronic arthritis, excluding gouty arthritis and reported them as arthritis.

Patients attended clinic visits at varying intervals determined by the doctors. Those with active comorbid conditions were seen more frequently while stable patients were seen less frequently. The longest interval between visits was 90 days. Patients who missed visits were followed up by a phone call which was recorded in the clinic's database. Patients without an attended visit for more than 90 days were identified using a system generated report and followed up by a phone call, or a home visit if the phone call follow-up was unsuccessful. When the patient could not be contacted through these strategies, they would be recorded as being lost to follow up.

We used frequencies, medians, and interquartile ranges to describe the cohort. The Wilcoxon Rank-Sum test was used to assess differences in skewed continuous variables across categories, and ordinary least-squares regression was used to determine a change in continuous variables over time. The square-root of CD4 T-cell counts was used to normalize the data, and regression analysis was used to compare changes in CD4 T-cell counts for each year on ART. The Chi-square test was used to compare categorical variables. We used Cox Proportional Hazards models to assess risk factors of death. Potential risk factors analyzed were sex, age at

enrollment, being married, being unemployed, having attended secondary or higher education, year of recruitment, having a WHO clinical stage 3 or 4 condition at ART baseline, baseline CD4 T-cell count and being hypertensive at enrollment. Potential risk factors of death were individually analyzed in univariate analyses. We included all potentially relevant risk factors (sex, age at ART commencement, marital status, education, ART baseline WHO clinical stage, ART commencement period, and being hypertensive) in a multivariable Cox Proportional Hazards model to determine independent risk factors of death. In the multivariable model, CD4 counts were grouped as <100 (severely immune-suppressed), 100–199 (moderately immune-suppressed), and ≥200 (relatively immune-competent). We used Stata (version 13.1, College Station, TX, USA) for all statistical analyses.

## Results

During the study period, 420 patients aged 50 years or older were enrolled into care at Newlands Clinic, 241 (57.4%) being female. The median age at ART commencement was 55 years (IQR: 52–59). There was no difference in ART commencement age between males and females (p = 0.241). More men were married (n = 152, 84.9%) compared to 58 women (24.4%), while 125 (52.5%) women were widowed compared to 15 (8.4%) men. Seventy-eight (45.1%) men were formally employed compared to 61 (26.5%) women, while 161 women (70.0%) were unemployed compared to 75 (43.4%) men. A total of 123 men (68.7%) had secondary or tertiary education compared to 126 (52.3%) women. Most patients (59.9%) were classified in WHO clinical stage 1 or 2 at ART commencement (Table 1). Three patients had a malignancy at the time of ART commencement.

### Attrition

The 420 patients were followed up for a total of 2612 person-years. The median duration of follow up was 5.6 years (IQR: 2.4–9.9). At the time of analysis (30 April 2020), 300 (71.4%) patients were still in care. During follow-up, 88 patients died (21.0%), 17 (4.1%) were lost to follow up, and 15 (3.6%) were transferred to other facilities (Table 2). Among those that died, 29 (33.0%) died during the first six months after ART commencement, and 38 (43.2%) during the first year. There was no difference in proportion of males died (23.5%) compared to females (19.1%) (p = 0.276). The median age of patients in care at the time of analysis was 63.6 years (IQR: 58.6–67.0).

### Antiretroviral therapy

The median baseline CD4 at ART commencement was 158 cells/uL (IQR: 86–274). Females had a higher median baseline CD4 count of 171 (IQR: 89–291) compared to males (median: 144 cells/uL; IQR: 79–236; p = 0.04). Overall, baseline CD4 T-cell counts increased by 11.8 cells/uL per year (95% CI: 7.9–15.6, p<0.01), but the increase among females (15.4 cells/uL per year, 95% CI: 9.9–21.0) was higher than among males (7.0 cells/μL per year, 95% CI: 2.3–11.6) (Fig 1). One hundred and forty-eight patients had a baseline HIV viral load with a mean ± SD of 4.5 log copies/ml ± 1.3.

Most patients (n = 390, 92.9%) were initiated on a first-line ART regimen containing a non-nucleoside reverse transcriptase inhibitor (NNRTI) (efavirenz or nevirapine) and two nucleo(t)side reverse transcriptase inhibitors (NRTI). The top three starting regimens were tenofovir + lamivudine + efavirenz, stavudine + lamivudine + nevirapine, and zidovudine + lamivudine + nevirapine with 124 (29.5%), 95 (22.6%), and 91 (21.7%) patients, respectively.

Overall, 178 (42.4%) patients experienced at least one treatment adverse effect. The antiretrovirals most discontinued due to adverse events were nevirapine, stavudine, zidovudine and

**Table 1. Baseline characteristics of older people enrolling for ART at Newlands Clinic (N = 420).**

| Characteristic | Frequency (%)[‡] | | | p-Value (chi[2] test) |
|---|---|---|---|---|
| | **Female** | **Male** | **Total** | |
| | *n = 241 (57.4)* | *n = 179 (42.6)* | | |
| Age at ART commencement, median (IQR) | 54 (52–59) | 55 (52–60) | 55 (52–59) | 0.241[§] |
| Marital Status | | | | **<0.001** |
| Married | 58 (24.1) | 152 (84.9) | 210 (50.0) | |
| Single | 26 (10.8) | 4 (2.2) | 30 (7.1) | |
| Divorced | 29 (12.0) | 8 (4.5) | 37 (8.8) | |
| Widowed | 125 (51.9) | 15 (8.4) | 140 (33.3) | |
| Unknown | 3 (1.2) | 0 | 3 (0.7) | |
| Employment Status | | | | **<0.001** |
| Formally Employed | 61 (26.5) | 78 (45.1) | 139 (34.5) | |
| Self-employed | 3 (1.3) | 11 (6.4) | 14 (3.5) | |
| Retired | 5 (2.2) | 9 (5.2) | 14 (3.5) | |
| Unemployed | 161 (70.0) | 75 (43.4) | 236 (58.6) | |
| Education | | | | **0.007** |
| None | 31 (12.9) | 12 (6.7) | 43 (10.2) | |
| Primary | 84 (34.9) | 44 (24.6) | 128 (30.5) | |
| Secondary | 89 (36.9) | 86 (48.0) | 175 (41.7) | |
| Tertiary | 37 (15.4) | 37 (20.7) | 74 (17.6) | |
| WHO stage[†] | | | | 0.727 |
| 1 | 81 (33.9) | 51 (29.1) | 132 (31.9) | |
| 2 | 67 (28.0) | 49 (28.0) | 116 (28.0) | |
| 3 | 64 (26.8) | 52 (29.7) | 116 (28.0) | |
| 4 | 27 (11.3) | 23 (13.1) | 50 (12.1) | |
| CD4 T-cell count (cells/µL), median (IQR) | | | | **0.045** |
| Overall | 171 (89–291) | 144 (79–236) | 158 (86–275) | |
| 2004–2010 | 139 (70–198) | 120 (63–181) | 127 (67–191) | |
| 2011–2015 | 221 (97–323) | 181 (97–287) | 199 (97–305) | |
| 2016–2020 | 238 (122–485) | 124 (84–305) | 196 (102–393) | |
| Log viral load (copies/ml), mean± SD | 4.3 ± 1.3 | 4.8 ± 1.3 | 4.5 ± 1.3 | **0.031[£]** |
| Comorbidities | | | | |
| Diabetes | 8 (3.3) | 9 (5.0) | 17 (4.0) | 0.380 |
| Hypertension | 43 (17.8) | 18 (10.1) | 61 (14.5) | **0.025** |
| Chronic Cardiac Failure | 3 (1.2) | 0 | 3 (0.7) | - |
| Arthritis | 1 (0.4) | 2 (1.1) | 3 (0.7) | 0.398 |
| Malignancies | 2 (0.8) | 1 (0.6) | 3 (0.7) | 0.744 |
| Chronic Kidney Disease | 1 (0.4) | 2 (1.1) | 3 (0.7) | 0.398 |

[‡]Unless otherwise specified

[†]414 patients had a recorded baseline WHO stage

[£]148 patients had baseline HIV viral load, difference calculated by t-test

[§]Wilcoxon Rank-Sum test

tenofovir disoproxil fumarate (TDF). The top five overall adverse effects were peripheral sensory polyneuropathy comprising 46.1% (n = 82) of all adverse effects, renal toxicity (n = 26, 14.6%), lipodystrophy (n = 17, 9.6%), zidovudine induced anaemia (n = 12, 6.7%) and grade 3 or 4 liver toxicity (n = 12, 6.7%). Stavudine-induced peripheral neuropathy accounted for 34.8% of all adverse events (n = 62).

**Table 2. Retention in care and attrition of a cohort of older people enrolling for ART at Newlands Clinic.**

| Status | Frequency (%) | | |
|---|---|---|---|
| | Female (n = 241) | Male (n = 179) | Total (N = 420) |
| In care | 179 (74.3) | 121 (67.6) | 300 (71.4) |
| Deceased | 46 (19.1) | 42 (23.5) | 88 (21.0) |
| Lost to follow up | 9 (3.7) | 8 (4.5) | 17 (4.0) |
| Transferred | 7 (2.9) | 8 (4.5) | 15 (3.6) |

Overall, seven patients experienced treatment failure and were switched to second-line ART comprising of a ritonavir boosted protease inhibitor (atazanavir or lopinavir) and two NRTIs. None of the patients failed second line ART. Among the 300 patients in care at the time of analysis, 283 (94.3%) had suppressed viral loads (<50 copies/ml), while 10 (3.3%) had viral loads >1000 copies/ml and seven (2.3%) had low-level viremia (50–1000 copies/ml). Among those with viral loads >1000 copies/ml, 5/10 had been receiving ART for less than 24 weeks. Among those who died, 28 (31.8%) had been receiving ART for less than 24 weeks.

From a median baseline CD4 T-cell count (IQR) of 158 cells/µL (86–274), median CD4 T-cell counts gradually increased from years 1–5 of follow-up to 254 cells/µL (181–399), 305 cells/µL (221–452), 348 cells/µL (249–464), 353 cells/µL (236–465), and 374 cells/µL (273–522) (Fig 2).

## Comorbidities

During follow-up, the most common incident comorbidities were hypertension, arthritis (all types of chronic arthritis), and chronic kidney disease (CKD) with 112 (26.7%), 83 (19.8%) and 73 (17.4%) new diagnoses, respectively (Table 3). Among the 300 patients in care at the end of follow-up, 148 (49.3%) had a diagnosis of hypertension with more females (n = 103, 57.5%) being hypertensive than males (n = 45, 37.2%) (p = 0.003). Among patients in care, 26 (8.7%) had a most recent estimated glomerular filtration rate (eGFR) <60mL/min/1.73m$^2$,

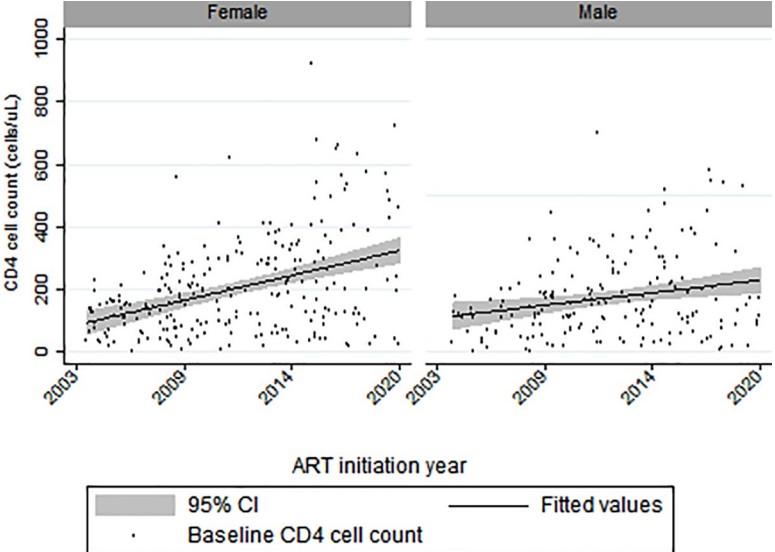

**Fig 1. Baseline CD4 count distribution over ART initiation years by sex (two records with CD4 count >1000 cells/µL omitted in the graph).**

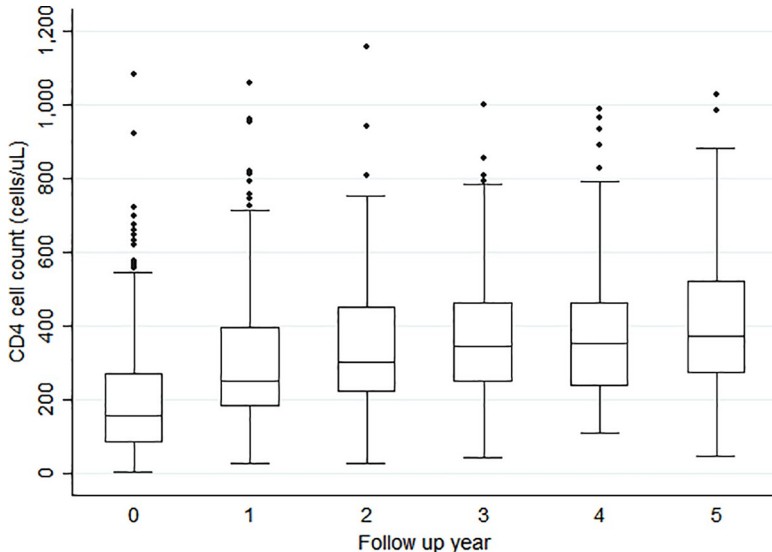

**Fig 2. CD4 T-cell count distribution across follow-up time (3 outliers, 2 Year 0 and 1 Year 1, with CD4 count >1250 cells/uL omitted from graph) (p<0.01).**

with 22 in CKD stage 3, and four in CKD stage 4. Of these 26 patients with CKD stage 3 or 4, 20 were female. Twenty-four (5.7%) patients, mostly female (18/26) had incident malignancies. Among females, eight were diagnosed with cervical carcinoma in situ III (stage zero cervical cancer), and five with cervical cancer. Other cancer types among women with one patient each were Kaposi's sarcoma, non-Hodgkin's lymphoma, myosarcoma, hepatoma and ocular malignancy not specified. Among males, two patients had Kaposi's sarcoma, two had multiple myeloma, one had cancer of the penis and another had lymphoma.

Overall, 259 (61.7%) of participants had at least one chronic non-communicable disease (NCD) whether prevalent at baseline or diagnosed during follow-up, excluding cervical carcinoma in situ III and vulvar intraepithelial neoplasia III. More females had NCDs (n = 163, 67.4%) than males (n = 96, 53.6%) (p < 0.004).

## Chronic disease medicines

Overall, 245 (58.3%) patients were receiving at least one additional chronic disease medicine other than ART (median = 2, IQR: 1–4). A higher proportion of females 156 (64.7%) were receiving additional chronic disease medicines compared to males (49.7%) (p = 0.002). Sixty-

**Table 3. Incident comorbidities among older patients receiving ART at Newlands Clinic (2004–2020).**

| Comorbid Condition | Frequency (%) | | |
|---|---|---|---|
| | Female (n = 241) | Male (n = 179) | Total (N = 420) |
| Tuberculosis | 26 (10.8) | 27 (15.1) | 53 (12.6) |
| Diabetes | 6 (2.5) | 7 (3.9) | 13 (3.1) |
| Hypertension | 71 (29.5) | 41 (22.9) | 112 (26.7) |
| Chronic Cardiac Failure | 7 (2.9) | 6 (3.4) | 13 (3.1) |
| Arthritis | 58 (24.1) | 25 (14.0) | 83 (19.8) |
| Chronic Kidney Disease | 42 (17.4) | 31 (17.3) | 73 (17.4) |
| Malignancy | 18 (7.5) | 6 (3.4) | 24 (5.7) |

**Table 4. Analysis of risk factors of all-cause mortality among older patients receiving ART at Newlands Clinic (2004–2020).**

| Characteristic | Hazard Ratio (95%CI) | *p* | Adjusted Hazard Ratio (95% CI) | *p* |
|---|---|---|---|---|
| Sex | | | | |
| Female | - | - | - | - |
| Male | 1.27 (0.84–1.93) | 0.262 | 2.29 (1.21–4.33) | **0.010** |
| ART commencement age | 1.04 (1.0–1.1) | 0.076 | 1.03 (0.99–1.07) | 0.200 |
| Married | - | - | - | - |
| Unmarried | 1.27 (0.83–1.93) | 0.273 | 2.06 (1.13–3.78) | **0.019** |
| Employed | - | - | - | - |
| Unemployed | 1.75 (1.08–2.85) | **0.024** | 2.01 (1.2–3.37) | **0.008** |
| Highest Education | | | | |
| Primary or none | - | - | - | - |
| Secondary or tertiary | 0.77 (0.51–1.18) | 0.230 | 0.83 (0.53–1.31) | 0.428 |
| ART Period | | | | |
| 2004–2010 | - | - | - | 0.242 |
| 2011–2015 | 1.20 (0.75–1.92) | 0.450 | 1.54 (0.93–2.56) | |
| 2016–2010 | 0.80 (0.38–1.69) | 0.558 | 1.09 (0.49–2.39) | |
| Baseline CD4 T-cell count (cells/μL) | | | | |
| <100 | - | - | - | **0.001** |
| 100–199 | 0.63 (0.39–1.02) | 0.058 | 0.72 (0.44–1.16) | |
| ≥200 | 0.38 (0.22–0.67) | **0.001** | 0.35 (0.19–0.63) | |
| Prevalent Hypertension | 0.59 (0.29–1.22) | 0.157 | 0.76 (0.36–1.6) | 0.464 |

Baseline WHO clinical stage was omitted from multivariable analysis due to collinearity with baseline CD4 T-cell count.

eight (16.2%) patients had more than three additional chronic disease medicines, and 24 (5.7%) (16 females and 8 males) were receiving more than five. The top five most used chronic disease medicines were the antihypertensives hydrochlorothiazide (16.1%), amlodipine (12.6%), nifedipine (12.6%), enalapril (9.7%), and atenolol (8.9%).

## Risk of death

In the univariate analysis, increasing enrolment age, unemployment, and baseline WHO status 3 or 4 (HR: 2.66 (1.72–4.12), p < 0.001) were significant predictors of death while increasing baseline CD4 T-cell counts were protective. Low CD4 T-cell counts, male sex, being unmarried and being unemployed were independent risk factors of death with male patients being twice as likely to die compared with females (Table 4). Patients who initiated ART with CD4 T-cell counts continued to have higher mortality rates throughout the follow-up period (Fig 3).

## Discussion

We present the treatment outcomes of OPLHIV who enrolled into care at Newlands Clinic in Harare, Zimbabwe, between February 2004, and April 2020. In this cohort, most patients were severely immune suppressed at enrolment, with significant differences between males and females in marital status, employment, and education. Hypertension was the most prevalent baseline and incident comorbidity, disproportionately affecting women. At analysis, 71.4% of patients were still in care, while 21% had died and only 4% had been lost to follow-up. ART related adverse effects were common, with 42% experiencing at least one side effect. ART was highly successful, with 94% of patients who had received ART for over six months being virally

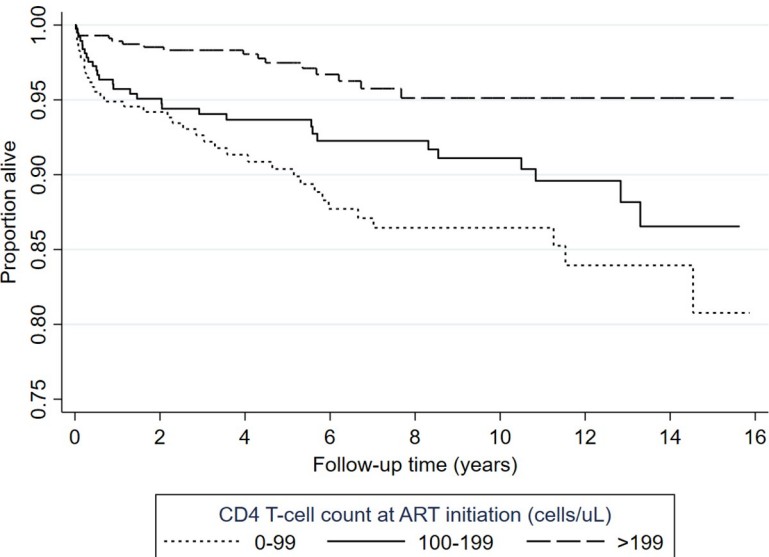

**Fig 3. Kaplan-Meier curve showing survivor estimates by baseline CD4 T-cell categories adjusted for sex, employment, and marital status.** Increasing baseline CD4 T-cell counts were associated with lower likelihood of failing ART (p = 0.017), whereas being male, unemployed, or unmarried were not associated with failing ART. Among patients still in care, the number of patients with unsuppressed viral loads was too low to assess for association between independent risk factors of death and having unsuppressed viral loads.

suppressed at the time of analysis. Over half of the patients (58%) received concomitant chronic disease medicines with their ART, mostly antihypertensives. Low baseline CD4 T-cell counts, being unmarried, being unemployed and being male were independent risk factors of all-cause mortality.

To our knowledge, this is the first study profiling outcomes of ART among OPLHIV in Zimbabwe. Our cohort was similar in sex distribution to other African cohorts that are comprised of more women than men [25–27]. Most of the women were widows in contrast to men who were mostly married. This difference is common in sub-Saharan Africa, where widows are less likely to remarry than widowers [28]. Older patients with HIV face a number of challenges that may affect their quality of life. Loss of partners and friends may lead to social isolation, depression and poor adherence to medicines [29]. The imbalance between men and women regarding education and employment is also reflected in this cohort, with more men having received education and being employed compared with women.

This study spans across a long period during which the WHO recommendations on CD4 T-cell count thresholds for ART commencement changed from <200 cells/μL to <350 cells/μL in 2010, then to <500 cells/μL in 2013, and finally "Treat-All" from 2015 [30, 31]. Nevertheless, the median baseline CD4 T-cell count at the start of ART only increased each year modestly, with a better improvement in baseline CD4 T-cell counts among women over time compared with males in line with findings from aggregated data from the IeDEA and the Collaboration of Observational HIV Epidemiological Research in Europe (COHERE) cohorts [32]. Nevertheless, many patients still presented with severe immunodeficiency and high viral loads (4.8 log copies/ml) across the study period [13]. This observation reinforces the call for interventions that target men for HIV testing and ART initiation.

Retention in care of 71.4% after a median follow up time of 5.6 years (IQR: 2.4–9.9) was comparable with that of a younger population (74.4%) at 36 months with a median age of 35 years (IQR: 28–42) in the Zimbabwe National ART Program and higher than that observed in a Kenyan cohort study (54%) at 36 months [33, 34]. However, unlike with this larger

Zimbabwean study of younger patients wherein 98% of attrition was due to loss to follow-up, 74% of attrition in our cohort was due to death. Retention in care was also higher and loss to follow-up lower than that from an aggregated analysis of data from IeDEA sites in Africa at five years of follow-up (66% retention in care and 18.8% LTFU in point adjusted analyses) [35]. Implementation of patient follow-up measures as employed by Newlands Clinic is more challenging for the National Program and other facilities that have much larger cohorts under their care.

Viral suppression was high amongst those patients in care at 94%. While there are conflicting data on viral suppression and immune recovery among older patients initiating ART [4], our cohort showed remarkable recovery during follow up with statistically significant increases in median CD4 T-cell counts with each year on ART. Stavudine has been discontinued as an ART option due to its high side effects profile that includes peripheral neuropathy, lipodystrophy, lactic acidosis and pancreatitis [36]. Peripheral sensory polyneuropathy was the most common ART adverse effect in this study. Older age has been shown to be associated with peripheral neuropathy in both HIV infected and uninfected people [37, 38]. Zidovudine and tenofovir disoproxil fumarate are still in use in LMIC and clinicians should continue to monitor elderly patients for adverse effects while on these medicines.

Prevalence of hypertension was high, as was its incidence during follow-up. This is consistent with previously published Zimbabwean data that estimated hypertension prevalence at 30% (95% CI: 19% - 42%) in the general population [18]. The prevalence of hypertension in this cohort at the end of follow-up (49.3%) was much higher than that of the general population, with a higher burden among females. Consequently, antihypertensives comprised the biggest proportion of additional medicines used by the participants in our cohort. Close to a fifth of patients developed CKD. This is also consistent with existing knowledge that the risk of CKD is high in PLHIV and increases with age [39]. Earlier studies in South Africa have also shown a high burden of hypertension, stroke, diabetes and obesity among OPLHIV [40]. Overall, women had a higher burden of NCDs compared with men. There are mixed findings on the distribution of NCDs among people living with HIV between men and women [41, 42]. Still, our findings are limited by our study design which used routinely collected clinical data and could not comprehensively assess for all important NCDs. Nevertheless, our findings reinforce the need for continuous hypertension and renal monitoring among OPLHIV.

Lower CD4 T-cell counts at ART commencement, being male, unmarried, or unemployed were independent risk factors of all-cause mortality in this cohort. Advanced immunosuppression, depicted by CD4 T-cell counts below 100 cells/μL, is an indicator of late ART initiation or presentation to care and is associated with high mortality across all age groups [33, 34, 43]. Male patients were twice as likely to die compared to women in our cohort in line with existing literature [13]. This difference could be explained by the higher viral loads and lower CD4 T-cell counts among male patients at ART initiation which are known to be associated with worse outcomes on ART [44]. Despite women being disproportionately affected by adverse social factors as well as hypertension, male patients still had a higher mortality risk in keeping with global general population trends [45]. The observed trend of persistently higher mortality throughout follow-up among people who initiated ART with lower CD4 T-cell counts and the lower baseline CD4 T-cell counts among men affirms the need for a shift in the HIV treatment strategy to improve the ART outcomes observed in men starting with enhanced enrolment strategies [15, 46]. The higher risk of mortality among unmarried people that we observed may be due to several factors including the effect of social relationships, including marriage, on mortality as well as a lack of treatment support [47, 48]. Unemployment was a significant risk factor of death in our cohort. Although employment alone is not a comprehensive wealth

index indicator, this finding is in line with a study conducted in Manicaland, Zimbabwe where a high wealth index was protective against both HIV infection and HIV-related mortality [49].

The strength of this study is the high quality of the data, which is ensured by routine data quality checks conducted by clinicians and a data quality manager. However, our study was based on routinely collected data, and we note some of the limitations associated with this method. HIV viral load monitoring was introduced in 2014, and analysis of viral suppression outcomes before this period could not be done. Cause of death information was not available in the records of deceased patients presenting a limitation in our analysis of risk factors of mortality. Furthermore, our study spans a 16-year period during which mortality risk factors may have evolved significantly. The outcomes of this cohort may not be generalizable to other facilities that have a less comprehensive approach to ART management than provided by Newlands Clinic or have much higher patient to clinician ratios. However, these outcomes show what can be achieved in the care of OPLHIV.

## Conclusion

Our study showed high HIV treatment success among OPLHIV, but with a high burden of hypertension, particularly among women. Low CD4 T-cell counts, male sex, being unmarried and being unemployed were independent predictors of mortality, highlighting the importance of social and socioeconomic factors in addition to the well described early ART for longevity among OPLHIV. We recommend psychosocial support programs to augment the "treat-all" approach aimed at providing ART early before severe immune suppression to improve longevity among OPLHIV.

## Supporting information

**S1 Checklist. The RECORD statement–checklist of items, extended from the STROBE statement, that should be reported in observational studies using routinely collected health data.**
(DOCX)

## Acknowledgments

We acknowledge the Newlands Clinic staff for contributing to the high-level quality data that were used in this study and members Institute of Social and Preventive Medicine for data analysis support.

## Author Contributions

**Conceptualization:** Tinei Shamu, Cleophas Chimbetete, Tinashe Mudzviti.

**Data curation:** Tinei Shamu, Tinashe Mudzviti.

**Formal analysis:** Tinei Shamu, Matthias Egger.

**Investigation:** Tinei Shamu, Tinashe Mudzviti.

**Methodology:** Tinei Shamu, Cleophas Chimbetete, Matthias Egger, Tinashe Mudzviti.

**Project administration:** Tinei Shamu.

**Supervision:** Matthias Egger, Tinashe Mudzviti.

**Validation:** Tinei Shamu, Cleophas Chimbetete, Matthias Egger, Tinashe Mudzviti.

**Visualization:** Tinei Shamu, Matthias Egger.

**Writing – original draft:** Tinei Shamu, Cleophas Chimbetete.

**Writing – review & editing:** Tinei Shamu, Cleophas Chimbetete, Matthias Egger, Tinashe Mudzviti.

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
