## [Decision Letter · Decision Letter 0]

19 Mar 2021

PONE-D-20-32848

Treatment outcomes in HIV infected patients older than 50 years attending an HIV clinic in Harare, Zimbabwe: A Cohort Study

PLOS ONE

Dear Dr. Shamu,

Thank you for submitting your manuscript to PLOS ONE. After careful consideration, we feel that it has merit but does not fully meet PLOS ONE’s publication criteria as it currently stands. Therefore, we invite you to submit a revised version of the manuscript that addresses the points raised during the review process.

We look forward to receiving your revised manuscript.

Kind regards,

Hyang Nina Kim, M.D., M.Sc.

Academic Editor

PLOS ONE

Additional Editor Comments:

In methods, please define chronic kidney disease.

Journal Requirements:

3. As part of your revision, please complete and submit a copy of the RECORD checklist, a document that aims to improve reporting and reproducibility of observational studies that use routinely-collected data for purposes of post-publication data analysis and reproducibility: (http://record-statement.org). Please include your completed checklist as a Supporting Information file. Note that if your paper is accepted for publication, this checklist will be published as part of your article.

Reviewers' comments:

Reviewer's Responses to Questions

**Comments to the Author**

1. Is the manuscript technically sound, and do the data support the conclusions?

Reviewer #1: Yes

Reviewer #2: Partly

2. Has the statistical analysis been performed appropriately and rigorously? 

Reviewer #1: Yes

Reviewer #2: Yes

3. Have the authors made all data underlying the findings in their manuscript fully available?

Reviewer #1: Yes

Reviewer #2: Yes

4. Is the manuscript presented in an intelligible fashion and written in standard English?

Reviewer #1: Yes

Reviewer #2: Yes

5. Review Comments to the Author

Reviewer #1: This well written manuscript describes HIV care and treatment outcomes as well as NCDs in OPLHIV starting HIV treatment aged 50 or older. Generally it is a well organized manuscript with clear methods, appropriate analysis, and sound discussion. I have only minor suggestions/questions for the authors:

1) For the definition of hypertension, on "two separate occasions" likely means two different days, as opposed to two measurements on the same day, but recommend clarification.

2) Recommend CD4 T cell count to CD4 cell count terminology

3) In the abstract, it appears that the number of men and women do not sum to the total studied, but then it becomes clear in the results that the numbers used in the abstract were just referring to the number with a given marital status. Consider showing the denominator of total n men and women in the abstract or another method to clarify.

4) In the introduction, it is stated that there is no similar data on OPLHIV in Zimbabwe - here would be a good place to comment briefly on the existence of OPLHIV data elsewhere in SSA, which is addressed in more detail in the discussion.

5) Recommend highlighting which were the adverse effects attributable to tenofovir, as this is a component the new standard first line regimen, TLD.

6) Would the study have captured aspirin use for risk reduction in those with HTN? Worth a comment, in my view.

7) the finding associated with increased risk of mortality and being unmarried merits some additional discussion.

Reviewer #2: This cohort study of 420 patients who are living with HIV and over 50 years old presenting for care and started ART at a clinic in Harare, Zimbabwe over a 16 year period. This cohort demonstrates a high mortality rate (21%) despite high rates of viral suppression (94%). High mortality correlated with delayed initiation of ART (WHO stage 3-4 at initiation) (aHR=3.0) and male sex (aHR=2.0) as well as indicators of poverty (unemployment) and being unmarried. I would recommend revising the analysis to include baseline CD4 count rather than WHO stage as a more objective measure of immunosuppression at initiation of ART. This manuscript is well written aside from some minor grammatical errors (e.g., inconsistent use of oxford comma) and includes interesting data that indicate the importance of considering gender and socioeconomic status when considering management of persons living with HIV who are >50 years old in order to reduce the risk of early mortality. This analysis adds to the evolving literature around aging and HIV and risk factors for mortality.

General comments:

1) Both male sex and increased years of uncontrolled viral load (low nadir CD4/high WHO stages) are independent risk factors for cardiovascular disease, and a high incidence rate of hypertension in this cohort, especially among female patients, is a notable mortality risk. However, the absence of cause of death data does not allow for further analysis and may reflect limited life expectancy for all adults in Zimbabwe. This should be more clearly indicated as a limitation of the analyses.

2) Given the long duration of 14 years over which this cohort was accrued, the authors should discuss the limitations of temporal trends that could have impacted mortality risks between 2004 and 2020.

3) Additional analyses may further illuminate the high mortality rate, especially since univariate analysis was not impacted by prevalent hypertension. In addition, differences by sex should be explored further by adjusting for CD4 count at treatment initiation. Longitudinal multivariate analysis of incidence hypertension, chronic kidney disease, malignancy, and TB would be needed to support the conclusions of this manuscript.

Specific comments:

1) Line 57 –Be more specific about special considerations for HIV care in aging adults living with HIV. Additionally, increased risk of opportunistic infections may not be true in the setting of viral suppression; please provide a reference and modify this comment as needed.

2) Lines 58-59 – Your data and the reference above indicate high rates of adherence in this demographic. Would you consider drug interactions altering drug availability or frequent changes in ART regimen to avoid drug interactions leading to disruptions more accurate that poor adherence?

3) Line 80- do you mean PLHIV (rather than OPLHIV) or does this clinic only provide care to patients over the age of 50 years old?

4) Line 84 – rather than using the term “vulnerable communities,” which implies weakness, UNAIDS terminology guidelines suggest using specific language such as “communities with high prevalence of poverty and unemployment.”

5) Line 132 – Clarify elderly, e.g., “420 patients 50 years or older…”

6) Line 152 – Given the elevated p-value, it would be appropriate to state that males and females died at a similar rate.

7) Line 202- If possible please clarify “cancer of the eye” or replace with “ocular malignancy not specified”

8) Risk of death analysis and Table 4

a) The baseline CD4 count was removed from the multivariate analysis because of collinearity with WHO stage. I would suggest removing WHO stage which is a more subjective measure than a CD4 count, especially given the apparent difference in CD4 count at initiation by sex (Figure 1) in the last decade.

b) Most participants provide multiple measures of blood pressure over time and the incidence of hypertension is quite high. It would be helpful to indicate what proportion of persons reported to have HTN were noted to have elevated BP at multiple visits.

c) Incidence of CKD, TB, and malignancy were also quite high. A longitudinal multivariate analysis of incident hypertension, CKD, TB, and malignancy correlation with mortality risk would be informative.

9) Lines 235-237, 251 – The authors note several differences between sexes in prevalence of social hardship and hypertension with females disproportionately impacted; however, male patients had a higher mortality rate. This warrants a comment from the authors.

10) Line 248 – Please clarify if you are referring to differences between males and females, sex, vs. social roles, genders. The word gender should be replaced with sex in this context.

11) Line 262 – Is this an observation from this cohort? Please ensure that these data on difference in baseline CD4 count and delayed treatment by sex are included in your results section. If these data are not available for your cohort, the reference to Western Cape cohort should be limited to the introduction section.

12) Line 269 – It is not possible to know if patients who died would have stayed in care if they were still living, and thus your data do not support your claim that older patients are more likely to remain committed to care than younger people, especially when the amount of patient tracking and staff outreach may differ at these different clinics.

13) Line 280 – Please comment on the prevalence of peripheral neuropathy and other side effects experienced in your cohort compared to other, younger cohorts.

14) Line 301-303 – Please explain your recommendation for enhancing enrollment strategies for males as a mechanism for reducing mortality among men living with HIV. It is curious that males had high rates of viral suppression but also had a high mortality rate. Figure 1 appears to show lower CD4 at initiation of ART, especially after 2011. Do you have additional data to explain higher mortality rate among males? Can risk of mortality among males be explained by CD4 count at initiation of ART?

15) Line 303-304 – Did unmarried patients have higher likelihood of failing treatment or uncontrolled viral loads? Please provide further data to explain the correlation between unmarried status and mortality in your cohort.

16) Line 320 – Please list all independent predictors of mortality not just WHO stage and unemployment.

17) Conclusion. The results mention the impact of TB and CKD but these were not included in your multivariate (or univariate) analyses. Were these predictors of mortality?

6. PLOS authors have the option to publish the peer review history of their article (what does this mean?). If published, this will include your full peer review and any attached files.

Reviewer #1: **Yes: **Julie A. Ake

Reviewer #2: No

---

## [Author Response · Author response to Decision Letter 0]

3 May 2021

Response to Reviewers

Responses to the Editor

Response

We have ensured that our manuscript is consistent with PLOS ONE’s requirements.

Response

We have reviewed the reference list and have not found any retracted papers. We have also checked completeness of the references.

3. As part of your revision, please complete and submit a copy of the RECORD checklist, a document that aims to improve reporting and reproducibility of observational studies that use routinely-collected data for purposes of post-publication data analysis and reproducibility: (http://record-statement.org). Please include your completed checklist as a Supporting Information file. Note that if your paper is accepted for publication, this checklist will be published as part of your article.

Response

Thank you for this recommendation. We have uploaded a copy of the RECORD checklist in this re-submission.

Response

The data are currently stored at figshare.com available on the link https://doi.org/10.6084/m9.figshare.14525487. 

Reviewer 1

1) For the definition of hypertension, on "two separate occasions" likely means two different days, as opposed to two measurements on the same day, but recommend clarification.

Response

Thank you for this comment. We have reviewed the sentence to read “Diagnosis of hypertension was made when a patient’s systolic or diastolic blood pressure was above 140mm Hg or above 90mm Hg, respectively, on two or more days”

2) Recommend CD4 T cell count to CD4 cell count terminology.

Response

We have replaced “CD4 cell count” with CD4 T-cell count.

3) In the abstract, it appears that the number of men and women do not sum to the total studied, but then it becomes clear in the results that the numbers used in the abstract were just referring to the number with a given marital status. Consider showing the denominator of total n men and women in the abstract or another method to clarify.

Response

Thank you for noting the discrepancy. We have corrected the figures in both the abstract and Table 1. We have added the denominator in the abstract for clarity.

4) In the introduction, it is stated that there is no similar data on OPLHIV in Zimbabwe - here would be a good place to comment briefly on the existence of OPLHIV data elsewhere in SSA, which is addressed in more detail in the discussion.

Response

Thank you for this comment. We have added a comment on data from the region.

5) Recommend highlighting which were the adverse effects attributable to tenofovir, as this is a component the new standard first line regimen, TLD.

Response

In the absence of definitive diagnoses of renal impairment which often require kidney biopsies, we could not objectively attribute specific adverse effects to TDF. However, as part of routine clinical care, TDF was stopped with presumptive diagnosis of TDF adverse effects.

6) Would the study have captured aspirin use for risk reduction in those with HTN? Worth a comment, in my view.

Response

We have added the statement “Risk reduction of major adverse cardiovascular events using aspirin was only provided for patients with history of myocardial infarction”. These patients are very few.

7) The finding associated with increased risk of mortality and being unmarried merits some additional discussion.

Response

We have added the statement “Older patients with HIV face a number of challenges that may affect their quality of life. Loss of partners and friends may lead to social isolation, depression and poor adherence to medicines”.

Reviewer 2

General comments:

1. Both male sex and increased years of uncontrolled viral load (low nadir CD4/high WHO stages) are independent risk factors for cardiovascular disease, and a high incidence rate of hypertension in this cohort, especially among female patients, is a notable mortality risk. However, the absence of cause of death data does not allow for further analysis and may reflect limited life expectancy for all adults in Zimbabwe. This should be more clearly indicated as a limitation of the analyses.

Response

Thank you for this comment. We have added this limitation in the paragraph discussing strengths and weaknesses of the study as highlighted below:

“Cause of death information was not available in the records of deceased patients presenting a limitation in our analysis of risk factors of mortality. Furthermore, our study spans a 16-year period during which mortality risk factors may have evolved significantly.”

2. Given the long duration of 14 years over which this cohort was accrued, the authors should discuss the limitations of temporal trends that could have impacted mortality risks between 2004 and 2020.

Response

Thank you for this comment. We have also added this limitation in the paragraph discussing strengths and weaknesses of the study as highlighted above.

3. Additional analyses may further illuminate the high mortality rate, especially since univariate analysis was not impacted by prevalent hypertension. In addition, differences by sex should be explored further by adjusting for CD4 count at treatment initiation. Longitudinal multivariate analysis of incidence hypertension, chronic kidney disease, malignancy, and TB would be needed to support the conclusions of this manuscript.

Response

Thank you for this comment. We analysed the mortality differences by sex adjusting for CD4 count alone and the difference was not apparent. However, adjusting for the other confounders showed that indeed sex is an independent risk factor. Instead of performing additional multivariable analyses on hypertension incidence, CKD, malignancy, and TB, we reviewed the conclusion to reflect findings that were observed in the original analysis. The conclusion now reads as follows:

“Our study showed high HIV treatment success among OPLHIV, but with a high burden of hypertension, particularly among women. Low CD4 T-cell counts, male sex, being unmarried and being unemployed were independent predictors of mortality, highlighting the importance of social and socioeconomic factors in addition to the well described early ART for longevity among OPLHIV. We recommend psychosocial support programs to augment the “treat-all” approach aimed at providing ART early before severe immune suppression to improve longevity among OPLHIV.”

Specific comments:

1. Line 57 –Be more specific about special considerations for HIV care in aging adults living with HIV. Additionally, increased risk of opportunistic infections may not be true in the setting of viral suppression; please provide a reference and modify this comment as needed.

Response

Thank you for this comment. We have reviewed the sentence as follows, “. Management of OPLHIV requires a multidisciplinary team approach, taking into account the onset of non-communicable diseases, premature ageing, and an increased risk of drug-to-drug interactions and drug toxicities due to polypharmacy”.

2. Lines 58-59 – Your data and the reference above indicate high rates of adherence in this demographic. Would you consider drug interactions altering drug availability or frequent changes in ART regimen to avoid drug interactions leading to disruptions more accurate that poor adherence?

Response

We agree with your comment. We have reconsidered and edited the sentence as follows: “Polypharmacy also complicates selection of ART regimens and the increased chronic pill burden may lead to treatment fatigue”.

3. Line 80- do you mean PLHIV (rather than OPLHIV) or does this clinic only provide care to patients over the age of 50 years old?

Response

Thank you for noting this error. We have corrected the sentence as follows:

“Apart from providing antiretroviral therapy to people living with HIV, the clinic also provides, among other services, onsite laboratory investigations, psychosocial support, reproductive health services including cervical cancer screening, dental services, food support, and bus fare assistance”.

4. Line 84 – rather than using the term “vulnerable communities,” which implies weakness, UNAIDS terminology guidelines suggest using specific language such as “communities with high prevalence of poverty and unemployment.”

Response

We have replaced “vulnerable communities” with the suggested terminology.

5. Line 132 – Clarify elderly, e.g., “420 patients 50 years or older…”

Response

We have removed the term “elderly” from this sentence and it now reads, “During the study period, 420 patients aged 50 years or older were enrolled into care at Newlands Clinic, 241 (57.4%) being female”

6. Line 152 – Given the elevated p-value, it would be appropriate to state that males and females died at a similar rate.

Response

Thank you for this observation. The sentence now reads, “There was no difference in proportion of males died (23.5%) compared to females (19.1%) (p = 0.276)”.

7. Line 202- If possible please clarify “cancer of the eye” or replace with “ocular malignancy not specified”

Response

Thank you for the suggestion. We have changed the term as suggested i.e., “ocular malignancy not specified”.

8. Risk of death analysis and Table 4

a) The baseline CD4 count was removed from the multivariate analysis because of collinearity with WHO stage. I would suggest removing WHO stage which is a more subjective measure than a CD4 count, especially given the apparent difference in CD4 count at initiation by sex (Figure 1) in the last decade.

Response

We have repeated the analysis with CD4 included and WHO stage excluded from the Cox model. Table 4 is now updated with new results.

b) Most participants provide multiple measures of blood pressure over time and the incidence of hypertension is quite high. It would be helpful to indicate what proportion of persons reported to have HTN were noted to have elevated BP at multiple visits.

Response

We agree that uncontrolled hypertension among hypertensive patients is interesting but would be better addressed as a different study. In the current study, we used the diagnosis variable to determine the burden of hypertension but we did not abstract individual blood pressure measurements from the database. As described in the methods section, a diagnosis of hypertension was made at the clinic when a patient’s systolic or diastolic blood pressure was above 140mm Hg or above 90mm Hg, respectively, on two or more days.

c) Incidence of CKD, TB, and malignancy were also quite high. A longitudinal multivariate analysis of incident hypertension, CKD, TB, and malignancy correlation with mortality risk would be informative.

Response

The authors agree with this comment. However, we feel that in-depth analysis of the various aspects listed above would go beyond the scope of this current study and may be better done as separate studies.

9. Lines 235-237, 251 – The authors note several differences between sexes in prevalence of social hardship and hypertension with females disproportionately impacted; however, male patients had a higher mortality rate. This warrants a comment from the authors.

Response

We have added the following statement in the discussion: “Despite women being disproportionately affected by adverse social factors as well as hypertension, male patients still had a higher mortality risk in keeping with global general population trends”.

10. Line 248 – Please clarify if you are referring to differences between males and females, sex, vs. social roles, genders. The word gender should be replaced with sex in this context.

Response

We have removed the word “gender” from the sentence, and it now reads, “This difference is common in sub-Saharan Africa, where widows are less likely to remarry than widowers.”

11. Line 262 – Is this an observation from this cohort? Please ensure that these data on difference in baseline CD4 count and delayed treatment by sex are included in your results section. If these data are not available for your cohort, the reference to Western Cape cohort should be limited to the introduction section.

Response

We agree with your comment. We have removed this sentence.

12. Line 269 – It is not possible to know if patients who died would have stayed in care if they were still living, and thus your data do not support your claim that older patients are more likely to remain committed to care than younger people, especially when the amount of patient tracking and staff outreach may differ at these different clinics.

Response

Thank you for this comment. We have edited the sentence to read as, “However, unlike with this larger Zimbabwean study of younger patients wherein 98% of attrition was due to loss to follow-up, 74% of attrition in our cohort was due to death.”

13. Line 280 – Please comment on the prevalence of peripheral neuropathy and other side effects experienced in your cohort compared to other, younger cohorts.

Response

We have added the following sentence in the discussion, “Older age has been shown to be associated with peripheral neuropathy in both HIV infected and uninfected people”.

14. Line 301-303 – Please explain your recommendation for enhancing enrollment strategies for males as a mechanism for reducing mortality among men living with HIV. It is curious that males had high rates of viral suppression but also had a high mortality rate. Figure 1 appears to show lower CD4 at initiation of ART, especially after 2011. Do you have additional data to explain higher mortality rate among males? Can risk of mortality among males be explained by CD4 count at initiation of ART?

Response

We have added Figure 4 which is depicts Kaplan-Meier survival estimates of mortality by baseline CD4 T-cell counts adjusted for sex, employment, and marital status. From the observed curves, we have edited the comment in the discussion as follows: “The observed trend of persistently higher mortality throughout the follow-up period among people who initiated ART with lower CD4 T-cell counts and the lower baseline CD4 T-cell counts among men affirms the need for a shift in the HIV treatment strategy to improve the ART outcomes observed in men starting with enhanced enrolment strategies”.

15. Line 303-304 – Did unmarried patients have higher likelihood of failing treatment or uncontrolled viral loads? Please provide further data to explain the correlation between unmarried status and mortality in your cohort.

Response

This comment has been partly addressed in comment number 7 under Reviewer 1. In addition, we have added the following statements after Table 4:

“Increasing baseline CD4 T-cell counts were associated with lower likelihood of failing ART (p = 0.017), whereas being male, unemployed, or unmarried were not associated with failing ART. Among patients still in care, the number of patients with unsuppressed viral loads was too low to assess for association between independent risk factors of death and having unsuppressed viral loads.”

16. Line 320 – Please list all independent predictors of mortality not just WHO stage and unemployment.

Response

Thank you for the comment. We have added all independent predictors of mortality to the conclusion.

17. Conclusion. The results mention the impact of TB and CKD but these were not included in your multivariate (or univariate) analyses. Were these predictors of mortality?

Response

Thank you for this comment. The conclusion has been reviewed as shown in the response to comment 3 under general comments.

---

## [Editor Report · Decision Letter 1]

18 May 2021

PONE-D-20-32848R1

Treatment outcomes in HIV infected patients older than 50 years attending an HIV clinic in Harare, Zimbabwe: A Cohort Study

PLOS ONE

Dear Dr. Shamu,

Thank you for submitting your manuscript to PLOS ONE. After careful consideration, we feel that it has merit but requires further clarifying revision. Therefore, we invite you to submit a revised version of the manuscript that addresses the points raised during the review process.

We look forward to receiving your revised manuscript.

Kind regards,

Hyang Nina Kim, M.D., M.Sc.

Academic Editor

PLOS ONE

Journal Requirements:

Additional Editor Comments (if provided):

(1) Please address the reviewer's comment re line 64 (formerly line 57) that discusses accelerated immune senescence and disease progression in the elderly. The reviewer point outs: "increased risk of opportunistic infections may not be true in the setting of viral suppression; please provide a reference and modify this comment as needed."

(2) Table 4 requires further revision to remove reference to WHO staging in favor of baseline CD4 cell count which is more objectively assessed than the former. There also remains a footnote re WHO staging that will need to be removed.

---

## [Author Response · Author response to Decision Letter 1]

24 May 2021

Thank you for the follow-up response to our re-submission. Below are responses to the two issues raised in the email dated 18 May 2021:

Comment 1 

Please address the reviewer's comment re line 64 (formerly line 57) that discusses accelerated immune senescence and disease progression in the elderly. The reviewer point outs: "increased risk of opportunistic infections may not be true in the setting of viral suppression; please provide a reference and modify this comment as needed."

Response

We previously revised the original sentence suggesting increased risk of opportunistic infections (OI) as suggested by the reviewer and removed the OI connotation. The revised sentence reads as follows, “Management of OPLHIV requires a multidisciplinary team approach, taking into account the onset of non-communicable diseases, premature ageing, and an increased risk of drug-to-drug interactions and drug toxicities due to polypharmacy”. To improve clarity, we have revised the preceding sentence as well which now reads, “Compounding the age-related thymic contraction, HIV infection affects both B- and T-cell function leading to accelerated immunosenescence and consequently increased risk of non-AIDS related, age associated comorbidities in elderly patients”, with the relevant references added.

Comment 2

Table 4 requires further revision to remove reference to WHO staging in favor of baseline CD4 cell count which is more objectively assessed than the former. There also remains a footnote re WHO staging that will need to be removed.

Response

We have removed the row with WHO stage from Table 4 and indicated the unadjusted hazard ratio in the text. We have also corrected the Table 4 footnote which now reads, “Baseline WHO clinical stage was omitted from multivariable analysis due to collinearity with baseline CD4 T-cell count.”

---

## [Editor Report · Decision Letter 2]

27 May 2021

Treatment outcomes in HIV infected patients older than 50 years attending an HIV clinic in Harare, Zimbabwe: A Cohort Study

PONE-D-20-32848R2

Dear Dr. Shamu,

We’re pleased to inform you that your manuscript has been judged scientifically suitable for publication and will be formally accepted for publication once it meets all outstanding technical requirements.

Kind regards,

Hyang Nina Kim, M.D., M.Sc.

Academic Editor

PLOS ONE

---

## [Editor Report · Acceptance letter]

1 Jun 2021

PONE-D-20-32848R2 

Treatment outcomes in HIV infected patients older than 50 years attending an HIV clinic in Harare, Zimbabwe: A cohort study 

Dear Dr. Shamu:

I'm pleased to inform you that your manuscript has been deemed suitable for publication in PLOS ONE. Congratulations! Your manuscript is now with our production department. 

Kind regards, 

on behalf of

Dr. Hyang Nina Kim 

Academic Editor

PLOS ONE